# High Job Burnout Predicts Low Heart Rate Variability in the Working Population after a First Episode of Acute Coronary Syndrome

**DOI:** 10.3390/ijerph18073431

**Published:** 2021-03-26

**Authors:** Yunke Shi, Ruxin Jiang, Caifeng Zhu, Min Zhang, Hongyan Cai, Zhao Hu, Yujia Ye, Yixi Liu, Huang Sun, Yiming Ma, Xingyu Cao, Dan Yang, Mingqiang Wang, Adrian Loerbroks, Jian Li

**Affiliations:** 1Cardiology Department, The First Affiliated Hospital of Kunming Medical University, Kunming 650032, China; shiyunke@ydyy.cn (Y.S.); caihy@ydyy.cn (H.C.); huzhao@ydyy.cn (Z.H.); yeyj@ydyy.cn (Y.Y.); liuyx@ydyy.cn (Y.L.); sunhuang@ydyy.cn (H.S.); maym@ydyy.cn (Y.M.); caoxy@ydyy.cn (X.C.); 2Cardiology Department, Baoshan People’s Hospital, Baoshan 678000, China; ruxinjiang2021@163.com; 3Cardiology Department, The People’s Hospital of Chuxiong Yi Autonomous Prefecture, Chuxiong 675000, China; Caifengzhu2021@163.com; 4Cardiology Department, Kunming Medical University, Kunming 650504, China; 15877917597@163.com (D.Y.); mingqiangwang2015@163.com (M.W.); 5Institute of Occupational, Social and Environmental Medicine, Centre for Health and Society, Faculty of Medicine, University of Düsseldorf, 40225 Düsseldorf, Germany; adrian.loerbroks@uni-duesseldorf.de; 6Department of Environmental Health Sciences, Fielding School of Public Health, School of Nursing, University of California Los Angeles, Los Angeles, CA 90095, USA; jianli2019@ucla.edu

**Keywords:** job burnout, acute coronary syndrome, working population, heart rate variability

## Abstract

(1) Background: Job burnout may affect the prognosis of patients with acute coronary syndrome (ACS) through mechanisms involving heart rate variability (HRV). However, no study has yet examined those potential associations. Hence, we conducted the present study to investigate this issue. (2) Method: Participants included patients who presented with a first episode of ACS and who were employed. The Copenhagen Burnout Inventory (CBI) was used to assess job burnout. Twenty-four-hour ambulatory electrocardiography recorded HRV on four occasions, i.e., during the hospitalization and follow-ups at one, six, and 12 months, respectively. (3) Results: A total of 120 participants who at least completed three Holter examinations throughout the study were enrolled in the final analysis. Job burnout scores at baseline were inversely associated with LnSDNN, LnTP, LnHF, LnLF, LnULF, and LnVLF during the consequent one-year follow-up. Each 1 SD increase in job burnout scores predicted a decline ranging from 0.10 to 0.47 in the parameters described above (all *p* < 0.05), and all relationships were independent of numerous confounders, including anxiety and depression. (4) Conclusion: High job burnout predicted reduced HRV parameters during the one-year period post-ACS in the working population.

## 1. Introduction

Acute coronary syndrome (ACS) remains the leading cause of death [1]. Survival rates of post-ACS have improved significantly with advancements in appropriate treatments, including early revascularization and evidence-based drug regimes [2]. Despite the observed decline in mortality and the improvement in survival rates after ACS, one-third of patients remain at a 20–30% risk of recurrent cardiovascular events during the subsequent 10 years [3,4]. Moreover, there has been a dramatic increase in the incidence of ACS among young people during the past decades [5]. A study evaluated data obtained from the Global Registry of Acute Coronary Events (GRACE) which showed that 6.2% of patients with ACS were younger than 45 years [6]. Young people are a critical resource for the labor market. However, potential ACS-related consequences (e.g., readmissions [7], heart failure [8], recurrent chest pain [9], or poor physical recovery [10]) and socioeconomic sequels (e.g., medical expenditure [11] and difficulties managing previous workloads [12]) adversely affected the social economy and increased the burden to society, families, and individuals. Thus, there is substantial room for improvement in terms of prognosis.

Over the past decades, the association between psychosocial factors and poor post-ACS prognosis has been examined extensively and has been confirmed independently of traditional risk factors, such as hypertension, diabetes mellitus, smoking, and hyperlipemia. Various mental disorders, such as depression, anxiety, or related comorbidities are common among those with ACS, and such disorders contribute to increased mortality and morbidity, and poor recovery after disease onset [13]. For instance, approximately 20% of patients with ACS in the hospital suffer from major depression, and the proportion of subclinical depression symptoms is even larger [14]. Moreover, robust associations between depression and a poor prognosis, including increased mortality and morbidity post-ACS, have been documented in various studies [15,16].

Being different from anxiety and depression, burnout is mainly characterized by emotional exhaustion [17,18]. In line with other types of mental status, burnout is associated with poorer cardiovascular outcomes [19]. Our previous study highlighted that burnout predicts a poor quality of life (QoL) and poor physical recovery among patients with a first episode of ACS [20]. The risk of recurrent events in patients with CHD seems to be elevated by 103% when patients experience vital exhaustion [21]. In recent years, studies have rarely focused on exploring the potential role of burnout in ACS.

The potential mechanisms linking burnout and ACS are complex and interrelated, and they include factors such as the metabolic syndrome, inflammation, immunity, and unhealthy behaviors. Various studies demonstrated that dysregulation of the hypothalamic-pituitary-adrenal (HPA) axis plays a significant role, resulting in adverse cardiovascular events. Ventricular arrhythmia is regarded as the most common factor that leads to sudden cardiac death, and the HPA axis is closely implicated in this condition, although the biological mechanism remains unclear [19]. Ventricular arrhythmia is difficult for clinicians to prevent, due to the lack of reliable risk stratification. Heart rate variability (HRV) is a non-invasive, convenient, and effective measure that estimates the functional balance between the sympathetic and parasympathetic nervous system, and it has been widely employed in clinical research [22]. Myocardial infarction (MI) induces dysregulation of the cardiac autonomic nerve, as evidenced by the presence of reduced HRV [23]. Furthermore, the decrease in the standard deviation of normal sinus RR intervals (SDNN) is associated with all-cause death in patients with post-MI [24]. Decreased HRV is associated with an increased incidence of major adverse cardiovascular events (MACE), mortality, and rehospitalization [25]. Thus, HRV is a potential predictive factor of risk stratification [26].

Our previous study showed that burnout was longitudinally associated with decreased HRV among patients after their first ACS during a one-year follow-up [27]. However, most of the participants in that study were retired people, and their working conditions were not taken into account. Therefore, general burnout (i.e., physical and psychological fatigue and exhaustion experienced by the person), rather than job burnout, was assessed in the previous study. Specially, job burnout is characterized as a physical and emotional exhaustion state due to prolonged exposure to work-related problems [28]. A few prior studies have examined associations between job burnout and HRV among healthy workers in the past years [29,30,31]. To the best of our knowledge, no longitudinal study has been carried out to date and no study has yet focused on individuals of the working population who present with a first episode of ACS. Therefore, we conducted a new longitudinal study to fill this research gap. Being different from the previous study mentioned above [27], this current study enrolled ACS patients who were still employed in the labor market. It aimed to examine the prospective associations of job burnout and HRV in working patients with ACS.

## 2. Materials and Methods

### 2.1. Study Participants

The participants included patients with a first episode of ACS who were still employed and who were admitted into the Cardiology Department of the First Affiliated Hospital of Kunming Medical University between March 2018 and December 2019, in Kunming, China. According to the definitions of clinical practice guidelines, ACS was characterized by unstable angina (UA), non-ST-segment elevation myocardial infarction (NSTEMI), and ST-segment elevation myocardial infarction (STEMI) [32,33]. Exclusion criteria included the following: (1) Frequency of arrhythmia affecting the HRV analysis. (2) Other comorbid conditions potentially interfering with symptom onset, or self-reported severe systemic disease, current use or a history of taking antipsychotic drugs. (3) Despite the help of others, patients were not able to understand or complete the questionnaire. (4) Patients were unwilling or unable to undergo the 24-h Holter ECG. All participants accepted coronary angiography during hospitalization. At one, six, and 12 months after discharge, patients returned back to the hospital for follow-up examinations. All participants provided their written informed consent. This project was approved by the Ethics Committee of Kunming Medical University (Kunming, China).

### 2.2. Collection of Sociodemographic and Medical Data

Data related to clinical profiles and demographics, such as age, sex, ACS type (UA, STEMI, NSTEMI), educational level (junior middle school or below, high school/technical secondary school, college, or higher), number of family members, monthly family income, medical history (hypertension, diabetes, hyperlipemia, stroke, smoking at present, heavy alcohol consumption), biochemical serum markers (glucose, total cholesterol, triglyceride, HDL-C, LDL-C), were collected from the medical record.

### 2.3. Assessment of Job Burnout, Anxiety, and Depression

Job burnout is a psychosocial factor, which is defined as a syndrome of emotional and physical exhaustion after long-term exposure to work-related problems [28]. Job burnout, anxiety, and depression were evaluated during hospitalization. Baseline measures were assessed using the Copenhagen Burnout Inventory (CBI)-Job Burnout Scale and the Hospital Anxiety and Depression Scale (HADS). Potential scores of the CBI ranged from 0 to 100, and a high score indicated high levels of job burnout. The reliability and validity of the Chinese version of the CBI were confirmed by other studies [34]. Our previous study also highlighted that the CBI was appropriate for Chinese patients with ACS to assess the burnout level [20,27]. All participants were assigned to a “high job burnout group” and a “low job burnout group,” according to the median point of the CBI job burnout score. Potential scores of the HADS ranged from 0 to 21. A score of ≥8 was taken to indicate anxiety or depression, and a high score indicated high levels of anxiety or depression [35]. In the present study, the Cronbach’s alpha coefficient for CBI job burnout subscale was 0.72, and the Cronbach’s alpha coefficient for the HADS-anxiety and HADS-depression subscale was 0.71 and 0.73, respectively.

### 2.4. Measurement of HRV

Four HRV examinations were performed throughout the study: before discharge (baseline), and one month, six months, and 12 months after discharge, respectively. The time for the first HRV examination was performed within one week after stabilization, and participants were informed to return to the hospital on schedule for the next 3 HRV examinations. HRV was measured by 24 h ambulatory electrocardiography beginning between 08:00–09:00 in the morning. A 12-channel Holter system, provided by Biomedical Systems in the United States, was used to record the continuous electrocardiograph (ECG) signal for 24 h for long-term measurements. HRV analysis was performed based on the Guidelines for Reporting Articles on Psychiatry and Heart rate variability (GRAPH): Recommendations to advance research communications [36]. The sampling rate of the hardware utilized was 500 Hz. The BMS Century 3000 HRV analysis software package (Version 2.0) was used to convert the analog ECG signal to a digital signal, which was analyzed after the recording. The standard deviation of all normal RR intervals (SDNN), which is one of the most common time-domain indices of HRV, was measured. Fast Fourier Transform was used to convert the normal 24-h RR interval into the heart rate power spectrum, and five frequency-domain measures were calculated [36]: (1) Total power (TP) had a frequency range of 0 to 0.4 Hz, which was the total power of the power spectral density, reflecting overall autonomic activity. (2) Ultra low frequency (ULF < 0.0033 Hz) power; (3) Very low frequency (VLF 0.0033 to 0.04 Hz) power; (4) Low frequency (LF), in a frequency range of 0.04 to 0.15 Hz, reflected the sympathetic activity that was modified by parasympathetic activity; (5) High frequency (HF), in the frequency range of 0.15 to 0.40 Hz reflected parasympathetic activity.

### 2.5. Statistical Analyses

Major demographic, clinical, and psychological data were summarized using either the mean and standard deviation, or frequencies, for continuous and categorical variables, respectively. The time-domain and frequency-domain parameters of HRV were transformed into their natural logarithmic values, i.e., ln(SDNN), ln(TP), ln(HF), ln(LF), ln(VLF), and ln(ULF), due to skewed distributions. Generalized estimating equations (GEEs) were used to analyze the association between job burnout at baseline and HRV parameters for repeated measurement during the 1-year follow-up. These analyses were adjusted for socio-demographic and clinical variables mentioned above, as well as anxiety and depression. In addition to the binary measures of job burnout (high vs. low), we also conducted sensitivity analyses with continuous measures of job burnout, and regression coefficients were reported for an increase by 1 standard deviation (SD). All data were analyzed using Stata version 10 software (Stata, College Station, TX, USA), and a two-tailed *p* value of <0.05 was considered statistically significant.

## 3. Results

### 3.1. Baseline Characteristics of Study Participants

A total of 123 patients agreed to participate in this study. Two of them were measured HRV less than three times during the study. One was allergic to ECG electrodes so failed to perform the 24-h Holter. Finally, 120 patients were included in this study who at least completed three Holter examinations throughout the study, of which 101 were male and 19 were female, whose mean age was 49.5 ± 7.6 years. The job burnout scores at baseline ranged from 8.30 to 95.80 with a median value of 50.00 points. We divided the participants into two groups: The patients with job burnout scores <50.00 points were assigned to the low job burnout group, while those with job burnout scores ≥50.00 points were allocated to the high job burnout group. The demographic and clinical data are shown in Table 1. Both the anxiety and depression scores of the high burnout group were higher than those of the low job burnout group, but only anxiety showed a statistically significant difference between the two groups. The incidence of hypertension and stroke in the high job burnout group was higher than that observed in the low job burnout group. However, the prevalence of family history of CVD showed an adverse trend between two groups. In respect to other data, there was no significant difference between the two groups.

### 3.2. Prospective Association of Job Burnout on HRV during Follow-Up

The HRV measurements at each of the four follow-ups are shown in Table 2. We observed a trend whereby the HRV parameters increased gradually during the recovery period.

Three GEE models were developed to detect the association between job burnout at baseline and the HRV parameters, which were repeatedly measured four times during the one-year follow-up. As shown in Table 3, compared with patients in the low job burnout group, the high job burnout level at baseline was consistently and inversely associated with all of the time- and frequency-domain HRV parameters, after adjusting for anxiety and depression, respectively. The results of the continuous job burnout scores were in line with the categorical variable described above (shown in Figure 1). Every increase of 1 SD in the job burnout scores corresponded to a different level of decline in the time-domain parameter (LnSDNN), which ranged from 0.10 to 0.11 (all *p* < 0.05), as well as to a decrease in the frequency-domain parameters (i.e., LnTP, LnHF, LnLF, LnULF, and LnVLF), which ranged from 0.21 to 0.47 (all *p* < 0.05), after adjusting for anxiety and depression, respectively.

Model I: adjustment for age, sex, ACS type (UA, STEMI, NSTEMI), educational level (junior middle school or below, high school/technical secondary school, college or higher), number of family members, monthly family income, medical history (hypertension, diabetes, hyperlipemia, stroke, family history of CVD, smoking at present, heavy alcohol consumption), biochemical serum markers (glucose, total cholesterol, triglyceride, HDL-C, LDL-C).

Model II: Model I + additional adjustment for anxiety.

Model III: Model I + additional adjustment for depression.

## 4. Discussion

Job burnout is a common negative physical and emotional response to prolonged exposure to work-related problems, and it is frequently observed among the working population exposed to the fast pace of today’s working life [28]. This study highlighted a significant negative association between job burnout and HRV, as reflected by time-domain and frequency-domain parameters, independent of anxiety, depression, and other traditional CVD risk factors. To the best of our knowledge, ours is the first longitudinal study to examine job burnout as a predictor of HRV in a working population after a first episode of ACS.

A decline in HRV has been observed in the acute phase after MI spanning across two weeks [37]. Previous studies reported that the decrease in SDNN, which was the most widely used index, was associated with poor cardiac outcomes [24,38]. Many studies have highlighted the inverse association between psychosocial factors and HRV in patients with CHD. Both depression and anxiety were associated with low HRV in post-MI patients [39]. Vital exhaustion was linked to reduced vagal function [30]. Job stress was found to be negatively associated with HRV [29]. A longitudinal study among apparently healthy employees provided evidence that job burnout was related to changes in HRV during a follow-up, independently of depression, suggesting HRV is a potential biophysiological mechanism underlying the link of job burnout with cardiovascular diseases [31]. Our previous study showed that higher general burnout scores relate to lower HRV in ACS patients [27]. Given the fact that the prevalence of ACS among the younger working population is increasing [5], this current study demonstrated consistent research findings that job burnout among employees with first episode of ACS was longitudinally associated with lower values for HRV parameters.

Large amounts of data have demonstrated a close association between a decline in SDNN and a worse prognosis following AMI. A meta-analysis revealed that the 3-year mortality of patients with SDNN values below 70 ms was four times higher than that of patients without significant SDNN reduction [38]. Furthermore, the incidence rate of sudden death was five times higher among patients with SDNN values below 50 ms [24]. The frequency-domain parameters, which included TP, ULF, and HF, were considered strong predictors of cardiovascular mortality in post-MI patients [25,26]. The negative association between job burnout and the HRV parameters in the present study indicated that job burnout was potentially linked to a poor prognosis in patients with ACS. HRV dynamically reflects the physiological perturbations of the autonomic nervous system function which is mediated by vagal and sympathetic nerve impulses [40]. Several studies demonstrated that HRV is sensitive to alternations in the automatic nerve system which is linked to stress [41]. Rizzo MR et al. showed a strong relationship between autonomic dysfunction and silent atrial fibrillation in type 2 diabetes [42]. The association between decreased HRV and low parasympathetic activity, a potential risk factor contributing to arrhythmia, is the most frequently reported in most studies. Garg et al. reported that vital exhaustion is associated with an increased risk of incident atrial fibrillation [43]. High levels of anxiety were associated with an increased risk of ventricular tachyarrhythmia [44]. However, the mechanism underlying the link between psychosocial factors, including job burnout, and adverse cardiac events is not fully understood. Further studies are required to investigate the relationship between job burnout, HRV, and prognosis in patients with ACS.

Our study has several strengths. First of all, this was the first longitudinal study to examine the association between job burnout and HRV in patients with ACS. A particular strength is that HRV was assessed as frequently as four times within one year. Second, compared with most post-studies which only enrolled patients with AMI as participants, our study included patients with UA, which is a more comprehensive type of ACS. However, our study also had some limitations. First, the sample size of this study was relatively small. Meanwhile, fewer female participants were included in our study, resulting in insufficient power to generalize the results of this study to women. Second, the methods that were utilized to assess psychosocial factors including job burnout, anxiety, and depression were based on self-administered questionnaires at baseline. Clinical interviews with validated instruments and repeated-measures of these psychosocial factors during follow-up would be preferred. Third, the HRV measurement may have been affected by multiple factors, such as the participant’s physical condition, exercise regime, breathing, and drug use, etc. Finally, as the follow-up period was limited to one year, we were not able to record long-term clinical outcomes such as death, major adverse cardiac events, and rehospitalization. A longer follow-up period would be beneficial to assess the association between job burnout and prognosis in patients with ACS.

## 5. Conclusions

In conclusion, our results demonstrated that high levels of job burnout were inversely associated with HRV in ACS patient across one year. The findings revealed that job burnout was a potential risk factor linked to poor prognosis in patients with ACS in long run.

## Figures and Tables

**Figure 1 ijerph-18-03431-f001:**
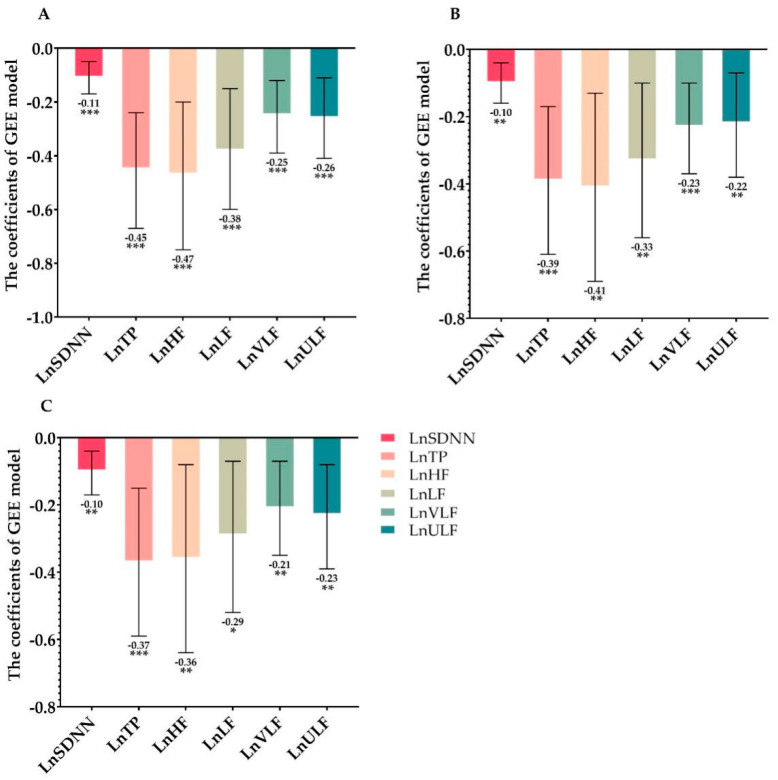
The coefficients (increase per SD) and 95% confidence interval of burnout scores at baseline with HRV parameters during one year using GEE models. (**A**) Model I: adjustment for age, sex, ACS type (UA, STEMI, NSTEMI), education level (junior middle school or below, high school/technical secondary school, college or higher), number of family members, monthly family income, medical history (hypertension, diabetes, hyperlipemia, stroke, family history of CVD, smoking at present, alcohol consumption), biochemical serum markers (glucose, total cholesterol, triglyceride, HDL-C, LDL-C). (**B**) Model II: Model I + additional adjustment for anxiety. (**C**) Model III: Model I + additional adjustment for depression. HRV: heart rate variability; Ln: natural logarithm; SDNN: standard deviation of NN intervals; TP: total power; HF: high frequency; LF: low frequency; VLF: very low frequency; ULF: ultra-low frequency; GEE: Generalized estimating equation. *. *p* < 0.05; **. *p* ≤ 0.005; ***. *p* ≤ 0.001.

**Table 1 ijerph-18-03431-t001:** Characteristics of study participants at baseline (*n* = 120).

		Low Job Burnout (*n* = 52)	High Job Burnout (*n* = 68)	*t*/*χ*^2^	*p*
Age (y)		48.75 ± 8.23	50.07 ± 7.00	−0.951	0.344
Sex [n(%)]					
	Male	47 (90.4)	54 (79.4)	2.662	0.103
	Female	5 (9.6)	14 (20.6)		
ACS type [n(%)]					
	UA	8 (15.4)	9 (13.2)	0.248	0.884
	STEMI	23 (44.2)	33 (48.5)		
	NSTEMI	21 (40.4)	26 (38.3)		
Education level [n(%)]					
	Junior middle school or below	11 (21.1)	23 (33.8)	5.118	0.077
	High school/technical secondary school	33 (63.5)	29 (42.6)		
	College or higher	8 (15.4)	16 (23.6)		
Number of family members (n)		4.00 ± 1.25	4.02 ± 1.37	−0.061	0.952
Monthly family income (Yuan)		7463.46 ± 2382.70	7095.59 ± 1952.61	0.929	0.355
Medical history (%)					
	Hypertension	13.3	28.3	4.484	0.034 *
	Diabetes	8.3	15.0	0.863	0.353
	Hyperlipemia	15.8	20.8	0.001	0.98
	Stroke	0.8	4.2	–	0.232 ^#^
	Family history of CVD	19.2	13.3	5.756	0.016 *
	Smoking at present	32.5	35.0	2.353	0.125
	Heavy drinker	15.8	16.7	0.682	0.409
Serum biochemistry (mmol/L)					
	Glucose	6.25 ± 2.29	6.58 ± 2.72	−0.700	0.485
	Total cholesterol	4.52 ± 1.27	4.38 ± 1.28	0.606	0.546
	Triglyceride	2.17 ± 1.13	1.79 ± 1.07	1.885	0.062
	HDL-C	1.17 ± 0.71	1.12 ± 0.40	0.549	0.584
	LDL-C	2.83 ± 1.05	2.89 ± 1.25	−0.294	0.769
Anxiety score		6.71 ± 3.79	8.29 ± 3.67	−2.311	0.023 *
Depression score		5.52 ± 2.00	6.28 ± 2.28	−1.910	0.059
Medication (%)					
	Aspirin	100	100	-	-
	P2Y12 receptor antagonists	100	100	-	-
	Statin	100	100	-	-
	Beta blockers	100	100	-	-
	ACEI or ARB	100	100	-	-

ACS: acute coronary syndrome; UA: unstable angina; STEMI: ST-segment elevated myocardial infarction; NSTEMI: non-ST-segment elevated myocardial infarction; CVD: cardiovascular disease; HDL-C: high-density lipoprotein cholesterol; LDL-C: low-density lipoprotein cholesterol; ACEI: angiotensin-converting enzyme inhibitors; ARB: angiotensin receptor blocker. *. <0.05; #. Fisher’s exact test.

**Table 2 ijerph-18-03431-t002:** The time- and frequency-domain HRV parameters during the follow-up periods.

HRV Parameters	Discharge	1 Month	6 Months	12 Months
Time domain HRV				
Ln(SDNN) (ln ms)	4.96 ± 4.67	4.99 ± 4.57	5.14 ± 4.31	5.15 ± 4.39
Frequency domain HRV				
Ln(TP) (ln ms^2^)	8.75 ± 1.97	9.23 ± 1.42	9.76 ± 1.27	10.23 ± 1.95
Ln(HF) (ln ms^2^)	7.60 ± 1.52	7.65 ± 1.85	8.31 ± 1.74	8.88 ± 1.04
Ln(LF) (ln ms^2^)	7.36 ± 1.27	7.53 ± 1.58	7.99 ± 1.47	8.19 ± 1.66
Ln(VLF) (ln ms^2^)	7.59 ± 0.98	7.83 ± 1.01	8.28 ± 0.86	8.56 ± 0.97
Ln(ULF) (ln ms^2^)	7.86 ± 0.97	8.01 ± 1.00	8.48 ± 1.04	9.34 ± 1.24

HRV: heart rate variability; SDNN: standard deviation of NN intervals; TP: total power; HF: high frequency; LF: low frequency; VLF: very low frequency; ULF: ultra-low frequency, Ln: natural logarithm.

**Table 3 ijerph-18-03431-t003:** The coefficients and 95% CIs of repeated measures of HRV parameters during follow-up by job burnout at baseline.

		Model I		Model II		Model III	
		Coefficients (95% CIs)	*p* Value	Coefficients (95% CIs)	*p* Value	Coefficients (95% CIs)	*p* Value
		LnSDNN					
Job burnout	Low	0.00		0.00		0.00	
	High	−0.19 (−0.31, −0.07)	0.002	−0.18 (−0.30, −0.05)	0.005	−0.18 (−0.30, −0.06)	0.004
		LnTP					
Job burnout	Low	0.00		0.00		0.00	
	High	−0.78 (−1.21, −0.36)	<0.001	−0.69 (−1.12, −0.26)	0.002	−0.68 (−1.09, −0.26)	0.002
		LnHF					
Job burnout	Low	0.00		0.00		0.00	
	High	−0.70 (−1.24, −0.15)	0.012	−0.60 (−1.13, −0.06)	0.029	−0.56 (−1.10, −0.03)	0.038
		LnLF					
Job burnout	Low	0.00		0.00		0.00	
	High	−0.54 (−0.99, −0.10)	0.016	−0.47 (−0.91, −0.03)	0.036	−0.44 (−0.88, −0.01)	0.044
		LnVLF					
Job burnout	Low	0.00		0.00		0.00	
	High	−0.43 (−0.69, −0.16)	0.002	−0.39 (−0.66, −0.13)	0.004	−0.37 (−0.63, −0.11)	0.005
		LnULF					
Job burnout	Low	0.00		0.00		0.00	
	High	−0.47 (−0.76, −0.18)	0.001	−0.42 (−0.71, −0.13)	0.005	−0.43 (−0.72, −0.14)	0.003

CIs: Coefficient intervals, Ln: natural logarithm.

## Data Availability

All data generated or analyzed during the study appear in the submitted article.

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
