# Peer review of "High Job Burnout Predicts Low Heart Rate Variability in the Working Population after a First Episode of Acute Coronary Syndrome"

_ijerph, 2021, doi:10.3390/ijerph18073431_

Round 1
Reviewer 1 Report
The work is interesting and original. The conclusions are supported by the results. The limitations of the study are quite indicated by the authors. The figures are clear.
This reviewer raises only a few criticisms that the authors need to address.
1- At baseline, both groups (low job burnout and high job burnout) are clearly biased in favor of the male gender. This aspect makes the results not generalizable to the female gender. This element should be included in the limitations of the study.
2- Moreover, at baseline there is a significant difference between the two groups in the family history of CVD. In model I and subsequent ones there is no adjustment for this parameter. This issue should be commented on in the discussion.
3- A correlation has been documented between alterations of the autonomic nervous system and episodes of stroke (Journal of Diabetes and its Complications, 2015, 29(1):88–92. doi:10.1016/j.jdiacomp.2014.09.002). This important topic needs to be commented on in the discussion.
Author Response
Response to Reviewer 1 Comments
Dear reviewer 1:
Thank you so much for your comments which are very important to improve our work. Accordingly, we have revised the manuscript based on your suggestions. Our point-by-point response is addressed below.
Point 1: At baseline, both groups (low job burnout and high job burnout) are clearly biased in favor of the male gender. This aspect makes the results not generalizable to the female gender. This element should be included in the limitations of the study.
Response 1: Given the fact of inclusion criteria of working population, the female is not the majority among ACS patients under 65 years old. In the revised manuscript, we added this point as one potential limitation. Please see Page 9, Line 294-295.
“Meanwhile, fewer female participants were included in our study, resulting in insufficient power to generalize the results of this study to women.”
Point 2: Moreover, at baseline there is a significant difference between the two groups in the family history of CVD. In model I and subsequent ones there is no adjustment for this parameter. This issue should be commented on in the discussion.
Response 2: Thank the reviewer for pointing this important issue out. We are deeply sorry for the error we made. Actually we did include family history of CVD as a confounding factor in the regression models, however, this variable was not mentioned due to our carelessness. We have made correction in the revised version. Please see Page 7, Line 225 and 235.
Point 3: A correlation has been documented between alterations of the autonomic nervous system and episodes of stroke (Journal of Diabetes and its Complications, 2015, 29(1):88–92. doi:10.1016/j.jdiacomp.2014.09.002). This important topic needs to be commented on in the discussion.
Response 3: This is a very good reference to strengthen our discussion. We have added this article as reference 42. Please see Page 8, Line 278-279; and Page 12, Line 440-442.
“Rizzo MR et al showed a strong relationship between autonomic dysfunction and silent atrial fibrillation in type 2 diabetes [42].”

Reviewer 2 Report
The authors aimed to examine the prospective associations of job burnout and heart rate variability (HRV) in 120 working patients with ACS. Job burnout scores at baseline were inversely associated with LnSDNN, LnTP, LnHF, LnLF, LnULF and LnVLF during the consequent one-year follow-up. Each 1 SD increase in job burnout scores predicted a decline ranging from 0.10 to 0.47 in the parameters described above (all p < 0.05). They concluded that high job burnout predicted reduced HRV parameters during the one-year period post-ACS in the working population.
This paper is very actual and clinically relevant. However, several issues should be considered to assess the results in this paper.
The authors mention that HRV is a potential predictive factor of risk stratification after ACS. The medication is very important for secondary prevention after ACS. However, they did not provide information about medication such as β-blocker etc. They should discuss the relationship between HRV and medication.
Author Response
Response to Reviewer 2 Comments
Dear reviewer 2:
Thank you so much for your comments which are very important to improve our work. Accordingly, we have revised the manuscript based on your suggestions. Our point-by-point response is addressed below.
Point 1: The authors mention that HRV is a potential predictive factor of risk stratification after ACS. The medication is very important for secondary prevention after ACS. However, they did not provide information about medication such as β-blocker etc. They should discuss the relationship between HRV and medication.
Response 1: It is great that the reviewer raised this issue of medication. In this study, all participants were treated with basic drugs for coronary heart disease after admission and during follow-up, including aspirin, P2Y12 receptor antagonists, statin, ACEI or ARB, and beta blockers. We did not display the medication information in the initial manuscript. Following your suggestion, we have added the medication in “Table1. Characteristics of study participants at baseline” on Page 5. As we can see, the role of confounding effect by medication could be ruled out because all patients across two groups took the same medication. Therefore, we did not discuss the relationship between medication and HRV. Well, it is indeed an important suggestion for our further research when recruiting new participants, we may be able to compare the influence of different dose of medication, such as beta blockers, on HRV in the future.

Reviewer 3 Report
The present study about the “High Job Burnout Predicts Low Heart Rate Variability in the 2 Working Population After a First Episode of the Acute Coro-3 nary Syndrome” is, interesting and providing the information how the High job burnout reduced HRV parameters in the working population after post-ACS. However, the authors should address the following points.
- Page1: 2nd line: please correct “survival rates of ACS”
- Authors are suggested to provide the details types of low job and high jobs [examples] and how they selected….. It would be beneficial if results are provided based on that
- In methods section Authors are suggested to provide the details of what exactly job burn out means, types of jobs how they considered low jobs and high jobs based on educational level or based on pay scale or based on stress levels. This is not clear. They have to explain in detail If possible, the results explained based on this job type would be more beneficial
- Similarly, in results section low job burn outs n-52 and the high job burn outs n=68. Did the authors find the similar results in equal number of comparison?
- Did the authors check for details of anxiety and depression in follow up of one year and please comment on how these factors might impact the ACS.
- Authors are suggested to proofread
Author Response
Response to Reviewer 3 Comments
Dear reviewer 3:
Thank you so much for your comments which are very important to improve our work. Accordingly, we have revised the manuscript based on your suggestions. Our point-by-point response is addressed below.
Point 1: Page1: 2nd line: please correct “survival rates of ACS”.
Response 1: The word “of” has been inserted after “survival rates”. Please see Page 1, last Line.
Point 2: Authors are suggested to provide the details types of low job and high jobs [examples] and how they selected….. It would be beneficial if results are provided based on that.
Response 2: In this study, the selection of low job burnout and high job burnout was not based on the job title. In fact, all participants were asked to fill in the Copenhagen Burnout Inventory (CBI)-Job Burnout Scale, and [All participants were assigned to a “high job burnout group” and a “low job burnout group”, according to the median point of the CBI job burnout score.] Please see Page 3, Line 133-135.
The participants in our study were employees with various jobs. Due to the relatively small sample size, the statistical power is quite limited to conduct detailed analyses by job titles. Indeed, we agree with the reviewer that job titles and occupational categories are crucial with respect to psychosocial work characteristics. When a large-scale study is planned in future, this issue will be definitely incorporated.
Point 3: In methods section Authors are suggested to provide the details of what exactly job burn out means, types of jobs how they considered low jobs and high jobs based on educational level or based on pay scale or based on stress levels. This is not clear. They have to explain in detail If possible, the results explained based on this job type would be more beneficial.
Response 3: As suggested, the definition of job burnout has been given in the revised manuscript. Please see Page 3, Line 125-126.
“Job burnout is a psychosocial factor, which is defined as a syndrome of emotional and physical exhaustion after long-term exposure to work-related problems [28].”
In this study, job burnout was measured by the Copenhagen Burnout Inventory (CBI)-Job Burnout Scale, which is a widely used and validated instrument for job burnout research. The details were described in “Methods” -- “Assessment of Job Burnout, Anxiety, and Depression”. Please see Page 3, Section 2.3.
Point 4: Similarly, in results section low job burn outs n=52 and the high job burn outs n=68. Did the authors find the similar results in equal number of comparison?
Response 4: Thanks for this question. In our analyses, we used the median point of the CBI job burnout score as the cut-off point to make two groups, low job burnout and high job burnout. Importantly, for purpose of minimizing the potential effect of arbitrary dichotomization of the independent variable (i.e., job burnout) and to fully using the data information, [In addition to the binary measures of job burnout (high vs. low), we also conducted sensitivity analyses with continuous measures of job burnout, and regression coefficients were reported for an increase by 1 standard deviation (SD).] Please see Page 4, Line 172-174.
[The results of the continuous job burnout scores were in line with the categorical variable described above (shown in Figure 1). Every increase of 1 SD in the job burnout scores corresponded to a different level of decline in the time-domain parameter (LnSDNN), which ranged from 0.10 to 0.11 (all p < 0.05), as well as to a decrease in the frequency-domain parameters (i.e., LnTP, LnHF, LnLF, LnULF, and LnVLF), which ranged from 0.21 to 0.47 (all p < 0.05), after adjusting for anxiety and depression, respectively.] Please see Page 6, Line 212-218; and Figure 1.
Point 5: Did the authors check for details of anxiety and depression in follow up of one year and please comment on how these factors might impact the ACS.
Response 5: Thank the reviewer for this important question. In the Introduction, we summarized that [Over the past decades, the association between psychosocial factors and poor post-ACS prognosis has been examined extensively and has been confirmed independently of traditional risk factors, such as hypertension, diabetes mellitus, smoking, and hyperlipemia. Various mental disorders, such as depression, anxiety, or related comorbidities are common among those with ACS, and such disorders contribute to increased mortality and morbidity, and poor recovery after disease onset.] Please see Page 2, second Paragraph; and reference 13-16.
It would be an ideal research design to include repeated measures of anxiety and depression. Unfortunately, anxiety and depression were only measured at baseline in our present study. In the revised manuscript, we added one limitation in the Discussion, please see Page 9, Line 295-299.
“Second, the methods that were utilized to assess psychosocial factors including job burnout, anxiety, and depression were based on self-administered questionnaires at baseline. Clinical interviews with validated instruments and repeated-measures of these psychosocial factors during follow-up would be preferred.”
Point 6: Authors are suggested to proofread.
Response 6: We have proofread our manuscript carefully, as suggested.

This manuscript is a resubmission of an earlier submission. The following is a list of the peer review reports and author responses from that submission.